# Social patterns underlying a new group formation in olive baboons

Jacob A. Feder[1,2]*, Shirley C. Strum[3,4], Joan B. Silk[1,2]

1 School of Human Evolution and Social Change, Tempe, Arizona, United States of America, 2 Institute of Human Origins, Arizona State University, Tempe, Arizona, United States of America, 3 School of Social Sciences and Department of Anthropology, University of California, San Diego, California, United States of America, 4 Uaso Ngiro Baboon Project, Laikipia, Kenya, East Africa

* jfeder2@asu.edu

## Abstract

In group-living animals, social and ecological challenges can push groups to fission into two or more 'daughter' groups. Here, we describe the demographic and social behavioral changes that were associated with the formation of a new group of olive baboons (*Papio anubis*) in Laikipia, Kenya. The process began when a high-ranking natal male transferred into a nearby study group, which coincided with the dispersal of several adult females. The dispersing females had close social ties with this male, and he had sired most of their current offspring. After a stint in the neighboring group, these animals eventually budded off to form a new, separate group. Throughout the fission process, female-female grooming was strongly predicted by eventual fission outcomes. In other words, females groomed most with the females they would remain with after the fission. By contrast, female-male grooming was prevalent in co-resident dyads but less strictly predicted by eventual fission outcomes. Although rates of aggression were elevated during periods when females dispersed, females who moved between groups were not targeted for eviction. Intergroup grooming remained elevated throughout the fission process, particularly between mixed-sex dyads, suggesting that group boundaries may have remained somewhat blurred. Taken together, the formation of this new group appears to have been a product of social factors including elevated levels of female-female aggression and females' affinity for particular males.

## Introduction

Primates living in large groups often experience costs stemming from heightened feeding competition [1–3], increased parasite loads [4, but see 5], or elevated infanticide risk [6,7]. In addition, dominant males in large groups are less able to monopolize mating opportunities [8–10]. To mitigate such costs, individuals residing in large groups may form temporary subgroups, as chimpanzees and spider monkeys

**Data availability statement:** Data and code are available at https://github.com/jacobafeder/Laikipia_Baboon_Fission and have been archived on Zenodo at https://doi.org/10.5281/zenodo.15398005.

**Funding:** This research was made possible due to funds from Arizona State University (to JBS) and the National Science Foundation (BCS-2313739 to JAF). These sponsors played no role in study design, data collection and analysis, decision to publish, or preparation of the manuscript.

**Competing interests:** The authors have declared that no competing interests exist.

frequently do [11], or initiate permanent fission, splitting to form two or more independent "daughter" groups [12].

The available data suggest that many primate fissions are triggered by female-female competition over food resources. In species that form very small groups, these female-initiated fissions can lead to the aggressive eviction of subordinate individuals, who occasionally go on to establish new social groups (e.g., lemurs: [13,14]). In species that form large, kin-structured societies, groups often divide along maternal kinship lines [15–19]. In these cases, social groups become increasingly cliquish in their grooming interactions [20,21], gradually establish separate home ranges, and eventually split into separate groups. By contrast, other primate fissions appear to be prompted by male-male mating competition. Male-initiated fissions may be less widespread than female-initiated fissions across primates, except perhaps in male-philopatric taxa (e.g., chimpanzees: [22,23]). In many male-initiated fissions, single males leave their social groups with a small set of females to form a new one-male group, often following alpha male replacements [24]. Such male-initiated fissions are common in Asian colobines and gorillas, which routinely form small, one-male groups [25–27], but have also been reported in baboons [28–31], which normally form larger, multi-male groups. When male-initiated fissions occur in baboons, female kinship ties can be broken if males aggressively herd females away from their maternal relatives.

Even within populations, fissions do not always follow the same pattern. Long-term studies of baboons and macaques, for instance, show that some fissions break maternal kin bonds while other fissions mainly preserve them [32,33]. In a population of Formosan macaques, some fissions followed a female-initiated, nepotistic pattern, while others followed a male-initiated pattern [34]. However, these sex-specific fission patterns are not mutually exclusive. For example, during a male-initiated fission, an inciting male could preserve female kin bonds by herding an entire matriline away from their natal group. Nevertheless, mutual benefits for females and males are not always certain. Fissions are often slow processes in which individuals may face conflicts when deciding which groupmates to follow, and dominant individuals may control the decisions of their groupmates, leading to seemingly idiosyncratic, often suboptimal outcomes [35,36].

Here, we describe the behavioral patterns that preceded and followed the formation of a new group of olive baboons (*Papio anubis*) living on the Laikipia Plateau of central Kenya. The fission process began when a male, Yohan, left his natal group and dispersed into a neighboring group. He was followed by a handful of adult females and their offspring during two waves of female dispersal. After several months, the male and the females that had followed him left this neighboring group and formed a new social group. Perhaps due to a deeper history of familiarity between these study groups, which we describe in further detail below, intergroup grooming persisted at high rates across the 4-year study period. This behavioral pattern, which is rare in baboons [37], presented us with the opportunity to examine whether the females' dispersal from PHG into ENK was preceded by heightened

intergroup grooming and identify the forces that encouraged prolonged intergroup interactions. We use data on grooming behavior and aggressive interactions to answer the following series of questions:

1. Were fission outcomes linked to the strength of pre-existing female-female or female-male social relationships?

2. How did grooming network structures change over the course of the dispersal and fission process?

3. Did female-female aggression shift in its frequency or intensity in the months prior to female dispersal events?

4. How did intergroup grooming relationships vary over the study period?

## Materials and methods

### Study population

The study was conducted on several groups of baboons monitored by the Uaso Ngiro Baboon Project (UNBP), directed by SCS [38]. The baboons occupy a savanna habitat and have wide, generalist diets. However, *Opuntia stricta*, an invasive fruiting cactus, has become widespread in recent decades [39] and comprised a large portion of the study groups' diets during the study period. This increased nutritional supply has supported faster reproductive rates, leading to the growth of both study groups and unhabituated census groups. This population growth has coincided with multiple group fissions [36]. From November 2013 to December 2017, detailed behavioral data were collected on two to three groups of olive baboons, supplementing UNBP's long-term data on these groups. One group, PHG, has been studied since 1971, and the second group, ENK, is the product of a previous fission from PHG that was completed in January 2011. The third group, YNT, was composed of individuals who initially dispersed from PHG to ENK and then left ENK to form a new group in May 2016.

Group membership in PHG, ENK, and YNT was assessed monthly using routine census records. At the end of each month, animals were retrospectively assigned to the group they were associated with during the majority of monthly observation days. We divided the dataset into four discrete periods bounded by the major demographic changes that occurred during the study period:

(1) *Stable* period (Nov 2013-Dec 2014), in which no adult females dispersed from PHG into ENK or vice versa. Yohan moved from PHG to ENK and back to PHG during this period.

(2) *Dispersal 1* period (Jan 2015–Sep 2015), which began when Yohan and three females moved from PHG to ENK; one of these females remained in ENK for the remainder of the study, and two of them resided in ENK for only 5 months and then returned to PHG; Yohan resided in ENK throughout this period.

(3) *Dispersal 2* period (Oct 2015–Apr 2016), which began when the two females that had returned to PHG moved back into ENK along with three other natal females from PHG; Yohan again resided in ENK throughout this period.

(4) *Fission* period (May 2016-Dec 2017), which began when Yohan and five of the six females that had dispersed from PHG to ENK formed a subgroup that ranged together and eventually became a fully independent group, YNT.

During the first two periods, there were approximately twice as many females in PHG as in ENK (mean ± 1 SD; PHG: 16.6 ± 1.6; ENK: 8.4 ± 1.5 females). During the Dispersal 2 period, there were approximately the same number of females in these two groups (PHG: 14.3 ± 1.0; ENK: 15.6 ± 0.8 females). In the Fission period there were again roughly twice as many females in PHG as in ENK, and only six females in the new group, YNT (PHG: 17.4 ± 1.6; ENK: 9.2 ± 0.4; YNT: 6.3 ± 0.5 females; **Fig 1**).

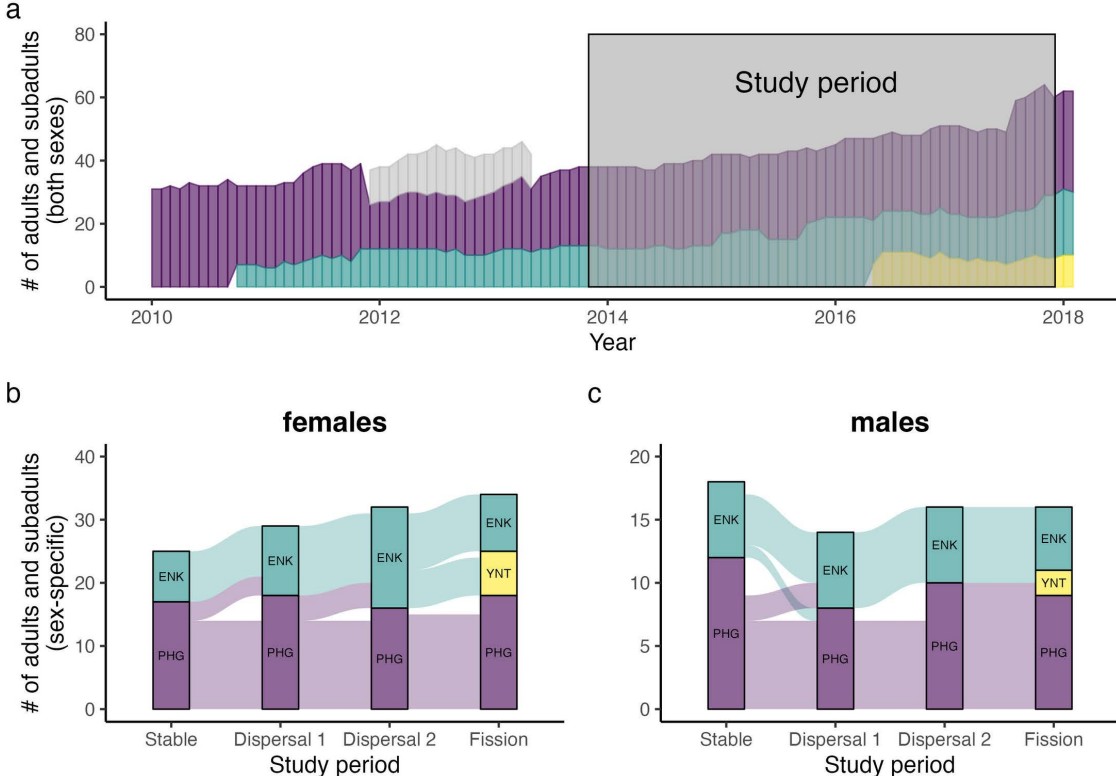

**Fig 1. Changes in demography across PHG and its fission products.** (a) indicates monthly shifts in group size across PHG and its fission products between 2010 and 2018. Gray bars indicate the fission product OGS, which was dropped from routine observations in 2013. We retain OGS individuals here to better visualize the shared history of these groups. The gray box indicates the duration of behavioral data collection for this study. (b) and (c) indicate more fine-grained changes in group composition for adult females and males and over the study period (via a Sankey diagram). At the start of the Dispersal 1 period, three females moved from PHG to ENK, although two briefly returned to PHG. These two females and three of their close relatives dispersed from PHG to ENK at the start of the Dispersal 2 period. These five females from PHG later left ENK to form YNT. Discrepancies between the height of bars across successive periods reflect (b) the net recruitment of females reaching adulthood or (c) the recruitment of natal males and the immigration of non-study males.

## Behavioral data collection

Behavioral data were collected during >14,000 15-minute focal animal observations between November 2013 and December 2017. We conducted focal observations on all females over the age of four (n = 31) and all males over the age of six (n = 21). This roughly corresponds to the ages at which females in our study groups begin sexual cycling and males begin dispersing from their natal groups.

During focal observations, observers recorded (1) the direction and duration of all grooming bouts and (2) the direction and sequence of all agonistic interactions among individuals within and between groups. Because of differences in the duration of the study periods, focal observation densities differed across the study periods (mean hrs/individual/period ± 1 SD: Stable = 22.6 ± 8.4; Dispersal 1 = 19.2 ± 4.6; Dispersal 2 = 12.2 ± 3.5; Fission = 22.3 ± 14.0 hrs/individual/period). Due to sampling priorities, observations of adult females were conducted about twice as often as observations of adult males (females: 22.8 ± 9.8; males: 12.1 ± 5.1 hrs/individual/period).

## Maternal relatedness

We assigned maternal relatedness using long-term demographic records. Genetically confirmed paternities and estimates of paternal relatedness were limited to more recent births and not known across the deeper population pedigree [40].

Thus, we assigned all female-female dyads a continuous maternal kinship value ($r_{maternal}$). Specifically, mother-offspring dyads were assigned 0.5, maternal sisters were assigned 0.25, and so on. Presumed non-kin dyads were given a 0. One unknown female that immigrated into the study population as a juvenile was assumed to be unrelated to all group members except her own offspring. Although this measure does not capture paternal relatedness, maternal relatedness is generally the more salient social force in baboons [41] and macaques [42, but see 43].

### Dominance ranks

Dominance ranks were assigned using the outcomes of all aggressive interactions within each troop (e.g., threats, chases, submissions) and applying a maximum likelihood approach to the Elo-rating method [44], in which dominance gains and losses are proportional to the probability of the individuals involved "winning" or "losing" a given interaction. Dominance ranks for adult females and adult males residing in the same study groups were calculated separately. However, female ranks were relatively stable across the study. Thus, to avoid computational noise during the 'burn-in' period, females were assigned fixed Elo-ratings for the entire study period. We then calculated relative ranks using the proportion of same-sex, co-resident groupmates each individual dominated in that month, which ranged from 0 to 1. In this way, ranks for both sexes shifted when group membership changed due to maturation, immigration, death, or fission, while only male ranks shifted in response to agonistic 'wins' and 'losses.' Absolute rank differences were calculated for each month and then averaged within each study period to assess the influence of rank differences on social interactions.

### Statistical analyses

**Q1: Dispersal patterns, group membership, and grooming relationships.** To quantify patterns of grooming across the study periods, we constructed two sets of Bayesian hurdle gamma models in 'brms' [45], assessing the undirected strength of female-female (Model 1a-c) and female-male (Model 1d-f) grooming relationships. Hurdle models are ideal for datasets like ours, which contain many zeros in the response variable (64.6% of 1521 female-female dyad-years, 78.2% of 1660 female-male dyad-years). Specifically, these models jointly assessed the factors that predicted (i) whether a dyad ever groomed (yes/no) and, if the dyad did groom, (ii) how much they groomed (in minutes). These models produced two estimates for each predictor: one for the presence/absence of grooming (i.e., the 'hurdle' component), and another for the duration of grooming (i.e., the 'gamma' component). Because observation time varied across periods, we included total dyadic sampling effort (i.e., the sum of their focal observation hours) as a logged offset term in the 'gamma' component. We also included multimembership random intercepts, which allow multiple individuals (i.e., both members of the dyad) to occupy the same random effect and thus are well-suited for dyadic data [46]. We did not use random slopes, as our primary categorical variables of interest (e.g., eventual outcome, sampling period) did not sufficiently vary within all sampled individuals. Across these models, parameter estimates for the hurdle and gamma portions of the model were directionally concordant. For brevity, we provide mostly gamma effect sizes within the main text, except for cases where only the 89% credible interval for the hurdle term did not overlap zero. Full model results are available in the *Supporting Information* (Tables S1-6 in S1 Appendix).

To best capture the fission process, we included all dyads across the study population, including both within- and between-group dyads. It might seem surprising that we included between-group dyads, as there is virtually no intergroup grooming in most baboon populations. Although ENK fissioned from PHG four years before our study began, the two groups continued to range in parallel, roost in nearby sleeping sites, and sometimes interacted peacefully. Thus, including these dyads allowed us to detect whether dispersing females engaged in "shopping" prior to their move.

To examine how current co-residence and eventual fission outcomes predicted grooming relationships (measured using the number of minutes the dyad groomed), we ran three models for each relationship type (i.e., female-female, female-male). First, we ran models (Model 1a, Model 1d) assessing the influence of <u>current</u> group membership (co-resident, yes/no) on grooming patterns. Second, we ran models assessing the influence of <u>eventual</u> outcome (Model 1b, Model 1e) on grooming patterns by assigning dyads to one of four categories:

(i)  *stayed together*, for dyads that were co-resident at the end of the study. While most of these dyads were co-resident at the beginning of the study as well, this category included dyads containing one female who permanently dispersed from PHG to ENK during Dispersal 1;

(ii)  *split apart*, for dyads that were co-resident in PHG or ENK during the Stable period but eventually ended up in different groups;

(iii)  *briefly together*, for dyads that were coresident in ENK during the Dispersal 1 or Dispersal 2 period, but eventually were separated when YNT formed;

(iv)  *never together*, for pairs that were not co-resident during any period of the study.

Third, we ran models assessing both co-residence and eventual outcome simultaneously (Model 1c, Model 1f). We then compared these models using leave-one-out cross-validation. We used this model selection approach, as current group membership and eventual social outcomes were highly correlated. In doing so, we quantified whether grooming patterns reflected co-residence or anticipated the eventual fission.

Kinship and female dominance rank are known to shape female-female grooming behavior in this population [47,48] and have been broadly linked with group membership decisions during other primate fissions [32,33]. Thus, for analyses of female-female grooming, we included maternal relatedness and rank differences as covariates (n = 464 unique female-female dyads). Male dominance rank and infant protection may also shape the outcome of male-initiated fissions [28–30]. Thus, in analyses of male-female grooming, we included the male's rank and whether the male had sired the female's most recent offspring as covariates (n = 562 unique male-female dyads). Analyses of male-female grooming excluded 54 mother-son and maternal sibling dyads because these pairs were unlikely to form mating relationships.

In order to determine whether dyads that split apart weakened their grooming ties in anticipation of their eventual fission or whether dispersing females engaged in shopping in the months before their departure, we also constructed models with interaction terms between study period and eventual outcome. However, models including this interaction term produced poorer model fits than models without the interaction term (leave-one-out cross-validation: $\Delta$elpd = −7.9 ± 5.8). Thus, we included sampling period as a main effect to account for population-wide changes in grooming rates. We report the output of these reduced models below.

**Q2: Shifts in grooming networks over the course of the fission process.**  To track changes in group-level grooming patterns throughout the study, we pooled posterior predictions from the female-female and female-male dyadic regression models (Models 1c,f including co-residence and eventual outcome to ensure comparability across these sex-specific estimates) to generate nine undirected, group-level grooming networks: one for each group during the Stable, Dispersal 1, and Dispersal 2 periods, and one for each of the three groups during the Fission period (PHG, ENK, YNT).

Here, each grooming relationship (i.e., 'edge weight') was calculated as the amount of time that dyad was expected to have groomed across 40 hours of observation (i.e., roughly the average dyadic sampling effort). Using 'igraph' [49], we calculated two measures of network structure: *density*, which reflects the proportion of dyads that were ever observed interacting, and *modularity* [50], which reflects how well the network can be sorted into clusters. To assign individuals to clusters, we used the Louvain clustering algorithm [51]. To track temporal patterns in the cohesion of our study groups, we generated 1000 posterior networks for each group-period and calculated 89% credible intervals to visualize uncertainty in our network metrics.

**Q3: Changes in aggression patterns across the fission process.**  To quantify patterns of female-female aggression throughout the fission process, we first modeled aggression at the individual-level, treating the number of agonistic events each female received as a Poisson outcome (Model 3a). We included dispersal status (i.e., whether the female dispersed during the study, yes/no), group stability (i.e., whether females in their social group dispersed at the start of the subsequent sampling period, yes/no), categorical group size (small: < 10 females, large: > 10 females), and sampling

period as covariates. This allowed us to determine whether dispersing females were targeted for 'eviction' from their social groups. We included group size as a categorical variable, as group sizes were discontinuous. We also included individual-level random intercepts in this model. Models examining aggression patterns at the dyad-level captured similar temporal trends (*Supporting Information*: Table S8 in S1 Appendix).

We also constructed a logistic model (Model 3b) to investigate factors that influenced whether or not a particular aggressive interaction involved physical contact (n = 2519 female-female interactions, n = 298 involving physical contact). Here, we included sampling period, eventual group outcome (i.e., stayed together, split apart, etc.), maternal relatedness, and rank difference (absolute value of actor rank – recipient rank) as fixed effects. Because this analysis was on a per-event basis, we did not include sampling effort as an offset term here. We again included actor and recipient as a multi-membership random intercept term.

We then sought to examine whether females in smaller groups were 'outranked' by females in larger groups [52,53], which would suggest a potential cost of forming a very small group via fission. To do so, we constructed a model (Model 3c) focusing solely on between-group interactions observed during focal observations (n = 393). Here, we modeled the probability that a focal female 'won' an agonistic interaction as a function of her relative group size (i.e., smaller or larger). We again included actor and recipient as a multimembership random intercept.

**Q4: Patterns of intergroup grooming.** Lastly, to assess the drivers of between-group grooming relationships, we constructed a zero-inflated binomial model (Model 4) examining the proportion of grooming interactions each individual directed outside versus within their social group across the study (43.3% of responses zero, indicating no intergroup grooming). For each individual-period, we tabulated the number of grooming interactions they engaged in with adult females and adult males within versus outside their current groups (between-group grooming: 601 interactions, 19.72 hours; within-group grooming: 5413 interactions, 173.35 hours of observation). Then, we modeled the proportion of grooming interactions they engaged in with individuals outside their social groups as a function of sampling period, dispersal status, and directional sex pairing (i.e., ff, fm, mf) as covariates. We excluded all bouts of male-male grooming, as these were exceedingly rare. We also included an interaction term between sampling period and dispersal status to determine whether the primary participants in between-group grooming shifted over the study. Actor identity and social group were included as random intercepts. In doing so, we quantified and compared the grooming budgets of each individual across the study.

All Bayesian models in these analyses were constructed in 'brms'. These models included weakly informative priors and were run for 4 chains with 4000 iterations each. We assessed model fits by inspecting trace plots and conducting posterior predictive checks. Rhat values were all <1.01, and bulk and tail effective sample sizes all exceeded 1000. The full outputs of all models are provided in the *Supporting Information* in S1 Appendix.

## Results

It is difficult to pinpoint the beginning and end of fission processes precisely. The series of events that led to the formation of YNT seems to have begun when Yohan moved from PHG to ENK in May 2014. He remained in ENK for three months and then moved back to PHG. In January 2015, Yohan returned to ENK along with three adult females from PHG. One of these females, whose mother resided in ENK, remained in ENK for the duration of the study. The other two females returned to PHG in June 2015, while Yohan remained in ENK. In October 2015, these two females moved back to ENK, along with two other adult females, two subadult females, and several dependent infants. A fifth adult female dispersed from PHG to ENK the following month, November 2015. Seven months later, in May 2016, Yohan and these adult and subadult females left ENK to form a new group, YNT.

### Q1: Grooming relationships reflected a mixture of current and future social conditions

As expected, females that lived in the same group groomed more than females who lived in different groups (Model 1a: $\beta_{g.co-resident}$ = 0.50, 89% CI = [0.30, 0.69], **Fig 2a**, $\beta_g$ indicating a parameter from the gamma component of the model).

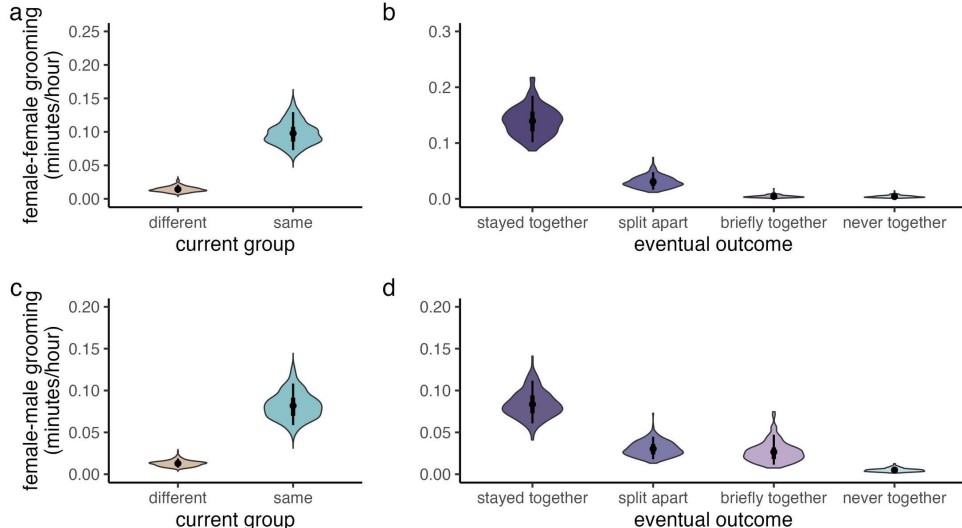

**Fig 2. On average, grooming relationships reflected current group membership (Models 1a, 1d) and anticipated eventual social outcomes (Models 1b, 1e).** Predicted grooming network edge weights, measured in minutes per observation hour, in relation to (a,c) current group co-residence and (b,d) eventual social outcomes. Violin plots provide the posterior distribution for each category of dyad. Thick and thin bars respectively indicate 50% and 89% credible intervals, while points indicate posterior means.

However, eventual fission outcome was a much better predictor of female-female grooming relationships than current co-residence (Model 1b vs. Model 1a: $\Delta$elpd = −99.3 ± 15.4). Specifically, female dyads that *stayed together* (reference category) groomed more than females who initially lived together but eventually *split apart* (Model 1b: $\beta_{g.split}$ = −0.50, 89% CI = [−0.73, −0.26]). Dyads that were only *briefly together* in ENK ($\beta_{g.briefly}$ = −0.84, 89% CI = [−1.25, −0.41]) and dyads were *never together* in the same group ($\beta_{g.never}$ = −0.99, 89% CI = [−1.33, −0.63]; **Fig 2b**) formed the weakest relationships. The model including both co-residence and eventual fission outcome (Model 1c) did not show improved fit over the model including eventual fission outcome alone (Model 1b vs. Model 1c: $\Delta$elpd = −1.5 ± 1.1). Thus, current co-residence did not capture variation in grooming patterns above and beyond that attributable to eventual outcome.

Throughout the study period, maternal kinship was linked with stronger grooming relationships (Model 1b: $\beta_{g.mat}$ = 0.21, 89% CI = [0.14, 0.28]), but absolute rank differences among females were not strongly linked to grooming ties ($\beta_{g.rank}$ = −0.07, 89% CI = [−0.16, 0.02]). To further assess the role of kinship in shaping fission outcomes, we constructed a *post hoc* Bernoulli model (Model S1) examining whether PHG females that stayed together were more likely to be maternal relatives than PHG females that split apart (i.e., maternal relatedness ≥0.0625, yes/no, 136 dyads). As expected, females that split apart were less likely to be maternal relatives than females that stayed together ($\beta_{split}$ = −1.39, 89% CI = [−2.36, −0.48]).

Female-male dyads residing in the same groups also formed stronger grooming relationships than female-male dyads residing in different groups (Model 1d: $\beta_{g.co-resident}$ = 0.54, 89% CI = [0.30, 0.77]; **Fig 2c**). Female-male dyads that *stayed together* throughout the study were also more likely to groom than female-male dyads that *split apart* (Model 1e: $\beta_{z.split}$ = 1.02, 89% CI = [0.71, 1.34]; **Fig 2d**, $\beta_z$ indicating a parameter from the 'hurdle' component of the model). For female-male dyads, models including both co-residence and eventual outcome performed better than models including co-residence or eventual outcome alone (Model 1f vs. Model 1d: $\Delta$elpd = −19.8 ± 6.9, Model 1f vs. Model 1e: $\Delta$elpd = −8.1 ± 4.0). When the effects of co-residence and eventual outcome were held constant, high-ranking males were also more likely to groom with females than low-ranking males (Model 1f: $\beta_{z.male.rank}$ = −0.37, 89% CI = [−0.64, −0.10], and females groomed more with males that had sired their most recent offspring ($\beta_{g.sire}$ = 0.42, 89% CI = [0.19, 0.65]).

This suggests that siring histories may have shaped female social decisions, and we examined past paternities to confirm this possibility. Yohan had sired six of the seven dependent offspring of the dispersing females (85%), but none of the eleven dependent offspring of females that remained in PHG. However, the broader influence of paternity on male group membership was limited, as sires did not always remain in the same groups as the mothers of their offspring. Excluding dyads that were terminated due to a male's disappearance, 64% (21 out of 33) of mother-sire pairs were co-resident at the conclusion of the study.

## Q2: Grooming networks shifted as the fission process progressed

Posterior grooming networks in PHG were less dense and more modular than grooming networks in ENK at the start of observation (Figs 3a, 3b). ENK became less dense and more modular during the Dispersal 1 period, and these patterns became even more pronounced in the Dispersal 2 period. This likely occurred because the dispersing females from PHG did not integrate well within the broader ENK network, resulting in the formation of discrete subgroups. Both PHG and ENK grooming networks became denser during the Fission period, when group membership stabilized. Unsurprisingly, the small fission product, YNT, was particularly dense and cohesive.

Although between-group grooming persisted throughout the study period, within-group grooming was far more prevalent (84.2% of female-female grooming, 83.8% of female-male grooming). Thus, grooming interactions largely matched our census-based group membership assignments, and the study population was clustered into two separate groups during the first three periods and into three groups in the last period (Fig 3c).

## Q3: Female-female aggression increased during periods of social instability

The frequency of female-female aggression varied across the study periods (reference category: Stable). Specifically, aggression increased during the Dispersal 1 period (Model 3a: $\beta_{dispersal.1} = 0.58$, 89% CI = [0.50, 0.65]) and then decreased

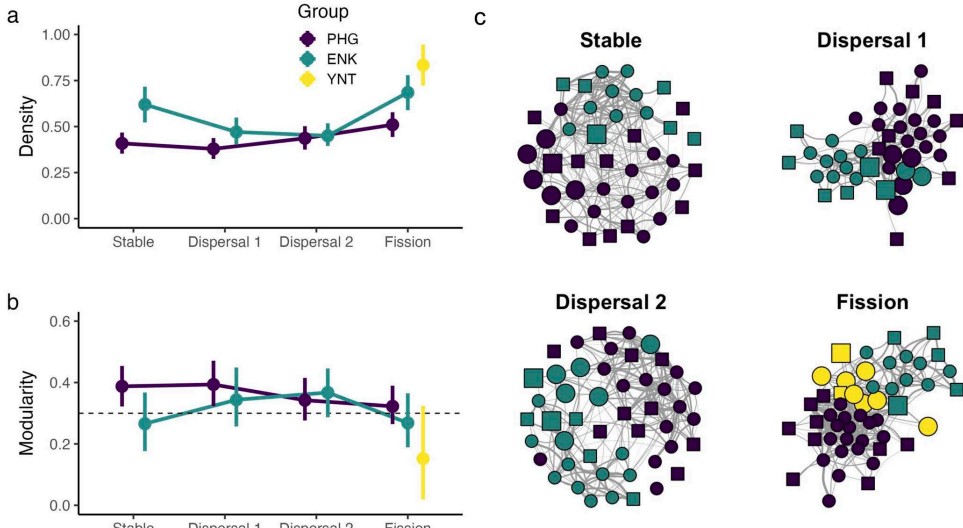

**Fig 3. Grooming networks shifted at the local group-level. (a,b)** Group-level trajectories in (a) density and (b) modularity across the four sampling periods. The dashed line in (b) indicates the conventional threshold at which networks are considered prone to fracture [41]. Points indicate median values from 1000 posterior networks, while bars indicate 89% posterior density intervals. **(c)** Community-wide grooming networks across the four sampling periods. The width of each edge corresponds to the associated dyad's edge weight. Nodes are colored depending on their current social group: purple indicates individuals that resided in PHG, teal indicates ENK, and yellow indicates YNT. Females are noted as circles and males as squares. Individuals that founded YNT are highlighted with larger vertices than all other nodes.

to a new low during the Fission period ($\beta_{fission}$=−0.41, 89% CI = [−0.49, −0.32]; **Fig 4a**). The females that dispersed from their natal group did not receive more aggression than permanent residents ($\beta_{dispersers}$=−0.18, 89% CI = [−0.42, −0.09]). Instead, aggression was broadly higher in unstable social groups, in which female dispersals were imminent ($\beta_{unstable}$=0.23, 89% CI = [0.13, 0.33]; **Fig 4b**). All else being equal, females in larger groups also tended to receive less aggression than females in smaller groups, although this pattern was generally weak and credibility intervals overlapped zero ($\beta_{large.group}$=−0.09, 89% CI = [−0.22, 0.03]).

The temporal patterns described above generally matched those for aggression intensity (reference category: Stable). Aggression was more likely to involve physical contact during the Dispersal 1 and Dispersal 2 periods (Model 3b: $\beta_{dispesal.1}$=0.47, 89% CI = [0.18, 0.76]; $\beta_{dispersal.2}$=0.87, 89% CI = [0.54, 1.20]), when group membership was in flux. Aggression was also slightly more likely to involve contact for dyads that split apart compared to dyads that stayed together ($\beta_{split}$=0.44, 89% CI = [0.06, 0.82]). Between-group aggression persisted at low levels throughout the study and was disproportionately directed from females in larger groups towards females in smaller groups (Model 3c: $\beta_{large.group}$ = 1.77, 89% CI = [1.35, 2.21]).

### Q4: Dispersers participated in more intergroup grooming than non-dispersers

Across the entire study period, grooming between groups comprised 17.6% of the total grooming budget. While between-group grooming was more common in dispersers than in non-dispersers, these differences were negligible during the Stable period (i.e., our reference category; Model 4: $\beta_{disperser}$=0.44, 89% CI = [−0.15, 1.01]) but widened over the course of the study, reaching a maximum difference during the Fission period ($\beta_{disperser \times fission}$=1.21, 89% CI = [0.88, 1.54]; **Fig 5a**). Between-group grooming was also disproportionately directed from females to males (reference category: female-female; $\beta_{fm}$=0.75, 89% CI = [0.52, 0.97]). More recent familiarity with outgroup individuals, facilitated by the dispersal of six natal PHG females and the concomitant movement of a handful of males, may have thus triggered increases in between-group grooming across the study period. Nevertheless, these high between-group grooming rates were not solely a byproduct of the current fission process, as between-group grooming was still appreciable during the Stable period.

## Discussion

In this study, we tracked the behavioral patterns that preceded and followed the formation of a new daughter baboon group. This process began when a male migrated from his natal group to a neighboring group. Dispersal is a normal part

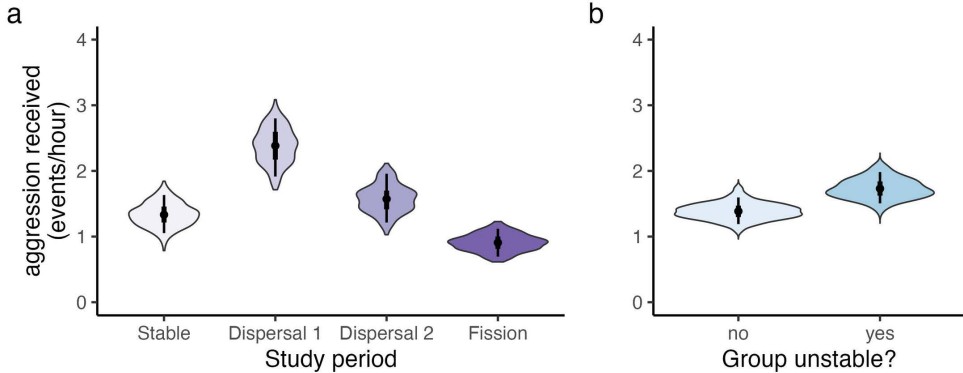

**Fig 4. Aggression rates rose during the dispersal periods and when social groups faced social instability (i.e., imminent dispersals). (a)** Aggression was elevated in Dispersal 1 and Dispersal 2 but declined to a new low during the Fission period. **(b)** Aggression was also elevated in unstable social groups. Violin plots provide the posterior distribution for each (a) sampling period and (b) dispersal category. Thick and thin bars respectively indicate 50% and 89% credible intervals, while points indicate posterior medians.

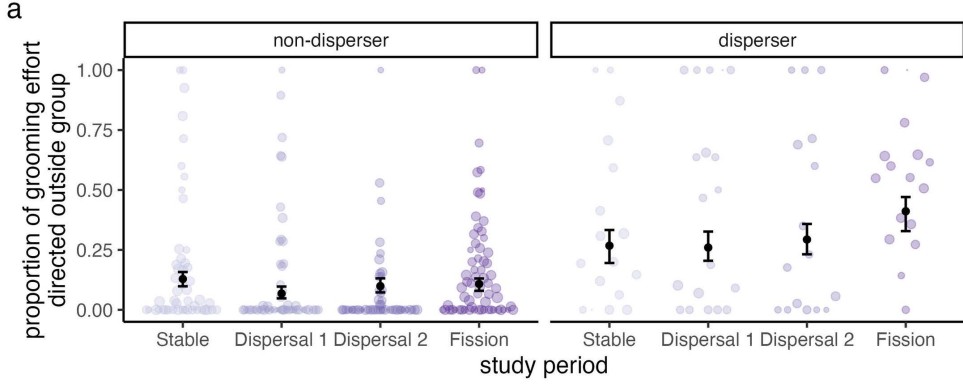

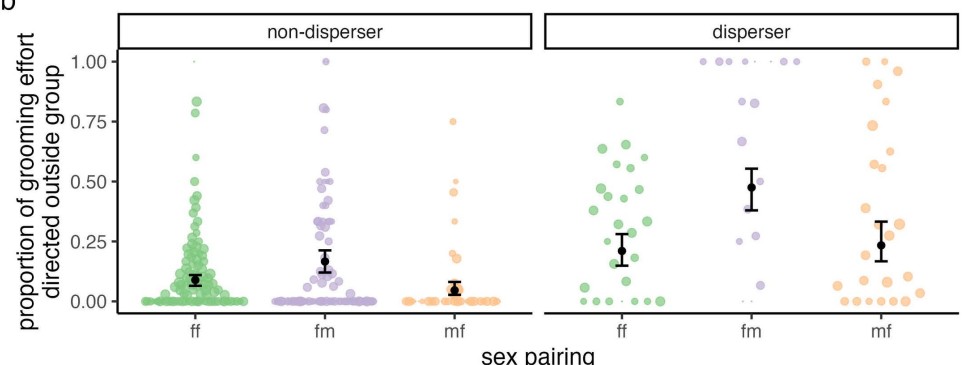

**Fig 5. Between-group grooming (a) increased after the dispersal event in dispersers but not in non-dispersers and (b) was most common among female-male dyads.** Data points indicate raw proportions of grooming effort directed outside each groomer's group across each sampling period. The size of each point is proportional to that individual's total grooming effort (in minutes) during a given sampling period. Points and bars indicate posterior means and 89% credible intervals.

of the life history of male baboons and reduces the risk of inbreeding [54,55]. However, this dispersal event was unusual because many of the dispersing males' closest female associates followed him into the neighboring group. After about a year, the male and his female associates budded off to form a new, independent group. Although social conditions in PHG, including a relatively high level of aggression and a more fragmented social network, may have played some role in females' decisions to leave PHG, the females who dispersed were not targeted for eviction. Instead, females' decisions about whether to stay or go seemed to be mainly influenced by their ties to the dispersing male and their relationships to one another.

## Social ties and maternal relatedness predicted patterns of fission

Females that stayed together in the same groups tended to show the strongest grooming relationships, while those that split apart or were briefly together tended to show weaker relationships. These patterns were even stronger than those linked to current group membership, particularly because the dispersing PHG females seldom groomed natal ENK females during their periods of co-residence. Thus, the fission process may have merely preserved preexisting social relationships. Previous studies of cercopithecine primate fissions have shown that kinship often predicts the composition of daughter groups, and fissions can provide an opportunity for females to increase their maternal relatedness to their group mates [15,32,33,56]. Likewise, in our study, the fission product largely consisted of females from a single matriline. The females in our study groups may have been strongly motivated to preserve their kin ties because females tend to form

their strongest, most enduring social bonds with close kin [41,57,58], and maintaining strong kin bonds has been linked to improved lifespan in many female-philopatric systems [59–62].

When deciding whether to stay or leave, females also appear to have been influenced by their ties to males, in particular the dispersing male YNT. Pregnant and lactating yellow, olive, and chacma baboons form close ties to particular adult males ("primary associates"), who are often the sires of their offspring [19,63–66]. These relationships are thought to provide baboon mothers and their offspring with a variety of benefits. In chacma baboons, males sometimes kill nursing infants after they enter new groups or rise to top-ranking positions, and primary associations may provide protection against infanticide [67]. By contrast, in yellow and olive baboons, in which infanticide is rare, primary associations may provide females and their infants with support in intragroup conflicts [68,69], preferential access to high-quality food resources [70], or protection against predation [71]. Immature baboons who co-reside longer and have closer ties to their fathers live substantially longer than their peers [72].

Thus, Yohan's departure from PHG created a potential dilemma for the mothers of his offspring – if they remained in PHG, they and their offspring would lose the benefits derived from their primary associations. By contrast, if they left PHG, they would sever important long-term ties to closely related females. However, because Yohan sired the offspring of several females from the same matriline, the conflict between maintaining their primary associations and kin ties was mostly avoided.

### Female-female aggression is unlikely to be the main factor underlying group fission

Previous studies of group fission in primates have found that feeding competition [15,17], injury rates [73,74], and mortality risk [23] may increase in the years leading up to and immediately following fissions. Female-female aggression rates in PHG nearly doubled in the Dispersal 1 period and were also high in ENK during the Dispersal 2 period, suggesting that female-female aggression might have played some role in females' dispersal decisions. It is not entirely clear why levels of female-female aggression increased during these two periods. However, in our study groups, females in small groups were in better condition than females in large groups, and high-ranking lactating females were in better condition than low-ranking lactating females [75]. In addition, rates of aggression by lactating females were higher than rates of aggression by females in other reproductive states. Taken together, these findings suggest that females face competition over access to food resources. However, other groups within the study population have maintained extremely large sizes (>150 individuals) for extended periods without fissioning. In addition, an earlier set of fissions in the study population appears to have been triggered by increases in the availability of anthropogenic foods, the spread of an invasive cactus, and subsequent shifts in foraging strategies [38,76]. Thus, it seems unlikely that heightened competition over resources in large groups was the primary factor underlying females' dispersal decisions in this population.

### Intergroup grooming persisted for a surprisingly long period

PHG and ENK, which completed their initial fission in 2011, continued to socialize, range in parallel, and co-sleep at nearby sleeping sites throughout the study period [36]. Similarly, the members of YNT continued to range near and interact with members of ENK and PHG after it budded off from ENK. Individuals that dispersed during the study (6 of the 31 females, 4 of the 21 males) engaged in more between-group grooming than individuals that did not switch groups during the study period. This may in large part reflect the fact that dispersers living in the new daughter group had a larger pool of familiar out-group partners to draw from. However, dispersal between groups can more broadly generate increased intergroup tolerance [77,78], and familiarity based on prior co-residence can even facilitate affiliative intergroup interactions (e.g., gorillas: [79], but not blue monkeys: [80]). Prolonged intergroup tolerance could be a byproduct of the site's ecological peculiarities, including an abundance of invasive fruits [39] and a scarcity of safe sleeping sites [38], which might encourage familiar groups to share space. As more detailed descriptions of the events preceding and following fissions

become available, we will develop a more complete understanding of the forces that shape fissions, their consequences for intergroup relationships, and the adaptive benefits that these demographic changes produce.

## Supporting information

**S1 Appendix. Supplemental figures and tables.** Additional figures and tables detailing our causal assumptions, posterior estimates from our statistical models, and model outputs.
(DOCX)

## Acknowledgments

We would like to thank the Office of the President of the Republic of Kenya and the Kenya Wildlife Service for permission to conduct this research. We would also like to thank Kate Abderholden, Megan Best, Megan Cole, Moira Donovan, Alexandra Duchesneau, Jessica Gunson, James King'au, Jeremiah Lendira, Joshua Lendira, Molly McEntee, Frances Molo, David Muiruri, Laura Peña, Eila Roberts and Leah Worthington for their assistance in data collection and field logistics at the UNBP. We thank Anja Widdig and three anonymous reviewers for useful feedback on previous versions of the paper.

## Author contributions

**Conceptualization:** Jacob A. Feder, Joan B. Silk.

**Data curation:** Jacob A. Feder, Joan B. Silk.

**Formal analysis:** Jacob A. Feder.

**Funding acquisition:** Jacob A. Feder, Joan B. Silk.

**Investigation:** Shirley C. Strum, Joan B. Silk.

**Methodology:** Jacob A. Feder, Joan B. Silk.

**Project administration:** Shirley C. Strum, Joan B. Silk.

**Resources:** Shirley C. Strum, Joan B. Silk.

**Supervision:** Joan B. Silk.

**Visualization:** Jacob A. Feder.

**Writing – original draft:** Jacob A. Feder, Joan B. Silk.

**Writing – review & editing:** Jacob A. Feder, Shirley C. Strum, Joan B. Silk.

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
