## [Decision Letter · Decision Letter 0]

28 Mar 2025

PLOS ONE

Dear Dr. Feder,

Thank you for submitting your manuscript to PLOS ONE. After careful consideration, we feel that it has merit but does not fully meet PLOS ONE’s publication criteria as it currently stands (details below). Therefore, we invite you to submit a revised version of the manuscript that addresses the points raised during the review process.

We look forward to receiving your revised manuscript.

Kind regards,

Anja Widdig

Academic Editor

PLOS ONE

Editor Comments

The manuscript has been reviewed by three experts and myself. The manuscript is well written and I enjoyed reading it. However, I asked a third reviewer to explicitly evaluate the statistical approach (given the complexity of the analysis). All reviewers raised several important concerns and I agree with them in all points. While the topic is really interesting and the data are sufficient to test the questions raised, the current version of the Ms. has serval serious limitations which I summarize below:

Introduction: would benefit from details on fission types observed in nonhuman primates (apart from male- and female-initiated splits). I think the main pattern are that splits occur across and between matrilines, while mainly females remain with close maternal relatives.Methods: most reviewer comments are related to the statistical models as they are hard to follow and often less clear. You run a total of eight Bayesian models and five frequentist analyses and it is unclear why you think they are necessary to address your questions. You nicely provide several questions, however, I recommend to link each question with one or more model/s. Furthermore, each analysis should be introduced separated while ideally being numbered (and/or list a short title), so the readers can view the methods and results per model in more detail. You can also think about moving some analytical parts of lower importance into the supplement to increase readability of the main Ms. In the current stage, the analysis seems to have several issues which need to be addressed in order to present it in a way readers can potentially digest it.Results: the results are difficult to follow. While they are linked to the broader question, they are not link to individual model/s. As pointed out by reviewers, please provide tables with model outcomes for all predictors and random effects including information on sample size and distribution of predictors.Conclusion: It is recommended to restrict conclusions to the actual results you are presenting.

In the light of these points, a major revision of the Manuscript is needed. If you are willing to provide a complete revision of the Ms, I will be happy to revise my decision and suggest the publication of this Ms. in Plos One. I think the study will be a valuable contribution to the literature.

P.S. Please submit the supplement with the revision.

Reviewers' comments:

Reviewer's Responses to Questions

**Comments to the Author**

1. Is the manuscript technically sound, and do the data support the conclusions?

Reviewer #1: Partly

Reviewer #2: Partly

Reviewer #3: Yes

2. Has the statistical analysis been performed appropriately and rigorously?

Reviewer #1: No

Reviewer #2: I Don't Know

Reviewer #3: N/A

3. Have the authors made all data underlying the findings in their manuscript fully available?

Reviewer #1: Yes

Reviewer #2: Yes

Reviewer #3: Yes

4. Is the manuscript presented in an intelligible fashion and written in standard English?

Reviewer #1: Yes

Reviewer #2: Yes

Reviewer #3: Yes

Reviewer #1: In this manuscript the authors describe changes in social behavior around a single group-fission event in a population of olive baboons. As a stand-alone case study the analyses will gain value only when others provide similar data fueling future statistical evaluation of the phenomenon. This being said, the manuscript provides a detailed account of the affiliation and aggression network before, during and after the fission event which is rare but desirable. The manuscript would benefit from a thorough revision of the intro where the two types of fission events observed in primates are defined and in the sequence of presentation of material. There is a glitch in the statistical analyses that needs fixing and discussion would benefit from relating the findings to a recent analyses of yellow baboon group fission events relative to female social bond networks. Below I provide comments as they came up during my first reading of the manuscript which mimics the experience of a future reader.

Introduction

Line 42: It may be noteworthy that in marginal habitats chacma baboons have been reported to form one-male groups and that some colleagues believe that these are the conditions during the speciation of chacma baboons that have left a trace in current day social behavior patterns and reproductive strategies including infanticide. As such, chacma baboons would fall within the same group as gorillas and Asian colobines – they have it in them to form one-male groups.

Line 51: The example does not really illustrate the point that individuals have different goals, because the male priority to recruit a number of females does not have to be at odds with females maintaining their close affiliative relationships if an entire matriline joins the event.

Line 56: The reader would benefit from a more detailed description of the published data on group splits in baboons and particularly those that are classified as male initiated. It is not entirely clear which criteria are used to classify events into male vs. female led. Is it the temporal pattern, a male goes first and the females join later? Is it a male actively herding females away from the main group? Is it always only one male or sometimes more than one? Is it not true that any number of females that start a new group will always do so together with one or several males?

Applied to the current study: What tells us that it was Yohan who led the females and not the females that led Yohan to ENK? Three females and Yohan went to ENK, 1 female and Yohan stayed. The 2 females that went back, retrieved another three to join ENK for a second stint and then decided to do their own thing for which they were joined by Yohan. The story could be told from the perspective of those two females that checked out opportunities, then mustered supporters to join ENK which failed and motivated these females to start a new group.

Methods

Kinship: It is not clear whether kinship was modeled as a continuous, dichotomous or a multinomial variable one and how this may have affected the results. On line 291 it seems that relatedness was classified into close kin versus non-kin, i.e. two categories? The data on github suggest a more fine-grained measure.

It seems that the statistical models do not include random slopes (e.g. the effect of group may vary across subjects or the effect of period may vary across subjects) and their correlations with random intercepts which can drastically increase type 1 error rate. Barr et al 2013 Random e�ects structure for con�rmatory hypothesis testing: Keep it maximal. J Memory and Language, 68:255–278; Schielzeth& Forstmeier 2009 Conclusions beyond support: overcon�dent estimates in mixed models. Behav Ecol 20:416–420.

Results

During my first reading I was not entirely sure about how females were assigned to a group. In Fig. 3 the estimated grooming time per hour for individuals that never stayed together clearly differs from zero and is just about ¼ of what you get if individuals lived in one group throughout. Where do you draw the line if there is so much grooming between members of different groups? This is even more extreme for between-sex grooming. Is that typical for the population, the species, baboons in general to see member of different groups grooming regularly? Why are all posterior estimates in the text negative?

As a reader I would prefer to see a table with model outcomes for all predictors and random effects including information on sample size and distribution of predictors (e.g. number of close kin and non-kin dyads).

Line 295: Something is missing here to indicate which estimate is for which of the groups compared. From the text is looks like dyads that stayed and those that separated were compared with some Wilcoxon Test. Since these dyadic data include repeated measurements the data points are not independent and we would look at a case of pseudo-replication inflating Type I error.

Line 296: This needs more explanation to make the analysis more accessible in the results section. How can there be as much grooming between females that currently live in the same and in different groups? Why are females grooming outside their group so much? Who is compared to whom here? The issue applies to analyses of female-male grooming and of aggression. Between-group grooming is mentioned in the methods section but appeared to be 10 times less frequent than within (600 vs. 5400 events).

Line 357: there is a word missing after slightly to indicate whether contact aggression was more or less likely.

Line 365: Perhaps you want to move this part about grooming between groups up to the beginning of the results section, because as you see from my comments, this amount of BG-grooming is unusual and very confusing to read about. The discussion talks about a priori knowledge on BG-grooming since 2011 which is info that could be moved all the way to the introduction.

Line 400: It is unclear how this test was run and how data were pseudo-replicated here as indicated by the very large W-value. This would become immediately clear if the sample sizes were reported with the test statistic and p-value.

Line 409: The conclusion that females were influenced by Yohan and their female kin seems a bit overstated. What is the evidence that rules out that females made their own decisions and Yohan and (and a bunch of subadult males) just joined the move? If as stated here, Yohan was natal to PNG, his natal dispersal had to come at some point anyways and thus may not have been driven by male-male competition but by incest avoidance.

The conclusion paragraph is redundant with the introduction and can be cut from the manuscript.

It would be great to discuss the results in relation to the Better baboon break-ups analyses by Lerch et al (a study cited here).

Reviewer #2: This was a detailed look at a group fission event, which provided insight into possible factors that influenced decision-making with regards of staying in a group, or leaving it. It is a nice manuscript to read, and I certainly feel I learned a lot about fissions and factors influencing who stays together.

However, I feel the paper would have benefited from more anecdotal information about the event, as this is primarily a case study of a fission. For example, what is missing is also more descriptive information about the initial dispersal of Yuhon. Once he dispersed, did the females immediately follow? Was there a brief period when the females stayed in their group (PHG) before following Yuhon? The female that stayed with Yuhon throughout Dispersal 1 and Dispersal 2, was there something different about her relationship to Yuhon or her female kin in PHG? What was different in the circumstances of the 1 female that did not remain with Yuhon?

The main take-home in the end seems to be that prior mating/siring history is what influenced the decisions of females to stick with Yuhon, and that these females were also closely related to each other. These are not captured strongly in all the many models in the paper, which makes me also wonder if the models were unnecessarily filled with too many fixed effects. Without any mention of how the different fixed effects influence each other (no DAGs), its less clear how to interpret each fixed effect estimate from each model. For example, maternal relatedness should definitely have a causal role in determining eventual outcome.

I feel this paper would have been easier to interpret had a model been focused on using data from the Stable period to predict the eventual outcome.

General comments:

I didn’t find any reference to the total number of focal subjects or the total number of dyads for the study. These numbers should be stated early in the Methods section (perhaps even the abstract), instead of the reader having to run code or look through SI information to find out.

I did not have access to the SI information, but please state in the main manuscript how model fit was assessed. Your SI material should include poster predictive checks, and trace plots to show good mixing of chains. What were considered acceptable rhat values and minimum effective sample sizes?

More specific comments:

Lines 84-87: Here would be the best place to introduce the information that intergroup- grooming continued to be common even after fissions. This did not come up until the discussion. This led to my confusion later (methods) as to why you would analyze data from dyads not co-resident in the same group.

Lines 120-122: “An additional four females dispersed from PHG to ENK at the start of the Dispersal 2 period”. I am trying to reconcile this with the information in lines 99-100 which states that “two females that had returned to PHG moved back into ENK along with three other females”. Isn’t this then “An additional three females”?

Line 125: the sub header “kinship” should be changed to “maternal relatedness” as the manuscript does not touch upon kinship relationships, only expected relatedness levels along the maternal line.

Lines 136: section on dominance. It is not clear why dominance has come up. It is not mentioned as a focus point in the introduction and suddenly appears here. Are there predictions for the influence of dominance rank? Or as it turns out rank difference in the models? The authors have not set up why it is important to consider in this study.

Line 151: Please include here the period over which these >14000 observations were taken. Yes, its in Figure 1, but it should also be mentioned here.

Somewhere in the section also please state the number of unique dyads in your dataset, broken down also by female-female, and female-male subsets.

Line 152: Please indicate 1) number of females that were focal-followed, and also the 2) number of males that were focal-followed..

Lines 166-180: If I am understanding this correctly, each ‘edge weight’ here is a point estimate. All the information about variation in sampling density is lost at this stage. The rest of this manuscript mentions Bayesian statistical approaches, so why not use Bayesian approaches here? With “nine networks for each behavioral type” (n=18 networks?), how much of the variance between networks in this manuscript is going to be due to noise because of differences in sampling effort. Figure 2 has no uncertainty around its point estimates.

Line 183-185: The authors state that they use a hurdle gamma model. 1) Please state explicitly that you do not use separate formulae for the hurdle and the gamma part. 2) You should give a brief description of what a hurdle gamma model is. Yhe average reader is not likely to know.

Lines 201-202, 222, 237-238: the authors will need to be explicit about what they mean by including “multimembership random effects term for dyad”. The average reader will have no idea what you mean by this. The model will return random intercept terms for Subjects, correct?

Statistical analyses:

This section was tough to get through on the first read through, given the various covariates and how they differed depending on the specific model. Then linking later results to specific models was time consuming. Please name each model in the statistics section, and bring back that model name in the dissemination of results.

State explicitly when you are using undirected data in your analyses. In most cases, you have aggregated by unique dyad (removing directionality of behaviour in most analyses). Though your data is directional in its raw form, it is no longer directional in most of your aggregated data tables that are fed into your models.

Please indicate what your reference categories are for analyses to make interpretation of results easier:

Outcome (4): stayedTogether, splitApart, BrieflyTogether, NeverTogether

Co-residence (2): Same/Different group

Sampling period (4): stable, dispersal1, dispersal2, fission

I struggled a bit when considering the relationship between co-residence and outcome. Am I correct in saying that dyads in the StayedTogether category are also always co-resident (Same category), and dyads in the NeverTogether category are also always not co-resident (Different category). So, co-residence only has variation in the splitApart and BrieflyTogether categories. So, really the interesting comparison would only be: do splitApart and BrieflyTogether dyads act differently when they are co-resident versus not (interaction).

Lines 215-216: this wording makes it unclear to me whether by “dyad-level” you mean 1) the total amount of aggression the dyad received in aggregate (sum of aggression to id1 and id2) or, 2) the total amount of aggression directed within the dyad (at each other, by each other).

Line 220: what is “current group combination”?

It is unclear to me why the individual-level version of the aggression model is necessary, as this is the same aggression data that is better represented in the dataset of the dyad-level dataset (an individual might at one extreme receive a lot of aggression from one individual, that drives their estimate up even though they receive little aggression overall from other baboons).

Figure 1b: I always love a good Sankey plot for this sort of information. As a service to the reader, perhaps mention what this type of graph is called, so that the reader can look up information on its use for visual representation of change in composition (in case they want to make one themselves in the future). It’s a very nice figure.

Results:

Lines 259-270: This section is hard to interpret because there is no indication of the strength of the evidence for these statements. As mentioned before, these appear to be based on point samples that discard uncertainty in their estimation due to variation in sampling density.

Lines 282-288: Unsure which analysis these results come from. The info is under “grooming networks” but sounds like the results are from the grooming hurdle gamma model. Where do these estimates come from? I think these are the estimates for Period in the grooming hurdle model. But I am confused because here it mentions that “we did not include these interaction terms in the models described below”. So, but these results do include an interaction? (so a separate model?). Please clarify.

Discussion:

Line 423: include here what counts as a close female relative (matR >= ?).

Reviewer #3: This manuscript describes and quantifies the demographic and behavioural dynamics during a group fusion and subsequent fission event in a population of olive baboons. The authors present an impressive data set, including affiliative and aggressive interactions of all adult animals in three social groups, collected over the course of several years. The group fission was initiated by a single adult high-ranking male, who first migrated to another group and later formed its own new group. Several females followed him. The authors found that those females had close social bonds with this male prior to his dispersal. Furthermore, these females had offspring sired by the dispersing male. The authors further found that the dispersing females were not evicted from the group, suggesting that dispersal was their active decision. Additionally, females tended to remain in the same social group as their closest grooming partners.

While the research questions are interesting and the data set to answer those is very comprehensive, I have a few concerns that should be addressed before publication. Please note that I was asked by the editor to specifically review the statistical analyses, rather than the whole manuscript.

Generally, I found it quite hard to follow the description of statistical analyses that have been conducted. To my understanding, the authors report a total of eight Bayesian models and five frequentist analyses. Especially in the results section I was struggling to understand from which model the reported numbers originated. That is why I would strongly suggest numbering the models consistently throughout the Methods and Results section, as was started in L214-215. Furthermore, the frequentist analyses came as a surprise to me in the Results and Discussion, as they have not been mentioned anywhere in the Methods. Please explain why and how you ran these tests in the appropriate sections.

L137-140: It is not clear to me if the ranks were calculated each month separately or if the ranks of the previous month were used as input to calculate the rank of the next month. Please clarify.

L143-144: Did female ranks remain stable, even if they changed group membership?

L151-154: Please mention how many females and males the data set includes in total.

Bayesian models: What is the reason you did not include random slopes in your models?

L185: Could you please indicate what percentage of the response values were zero?

L197-198: Would it not be interesting to see if kin biased social preferences play a role as well in the grooming dynamics over a fission event?

L200-201: How was the total dyadic sampling effort quantified?

L203-210: Curiosity question: Did the authors try to improve the model fit by increasing the iterations, delta and the maximum tree depth? It seems that the default settings of brms worked for all other models (L254-256). However, adapting the parameters could solve the convergence problems and enable them to answer these very interesting questions.

L213-238: The authors describe four models to test for aggression dynamics in females. Only one out of four models accounts for sampling effort and I wonder why this was not done for all four?

L218-219: Why was the group size included as a categorical measure rather than the actual size?

L220: What is “current group combination”? Does this measure indicate if the individuals are in the same or a different group? Or did the authors specifically include in which groups the individuals were?

L228-231: I wonder why the authors did not include the current group of residence as a fixed effect in the model, as the groups might differ in aggression dynamics. Please clarify.

L240-244: Why did the authors chose a zero-inflation rather than a hurdle model in this case? Also, it is not clear to me if the reported error distribution was applied to both models or just the latter. Please clarify and report what percentage of the response values were zero.

L282-319: The authors report that they performed hurdle models for the grooming data, as it contained an excess of zeroes (L183-185). A hurdle model consists of two parts: A hurdle and a count part. The hurdle part treats the response as binary variable and estimates which predictors influence that the response is unequal to zero. The count part of the model only takes observations into account where the response is unequal to zero and estimates which predictors influence the magnitude of the response. However, in the results section, the authors only report one estimate per predictor and do not specify if that originates from the hurdle or count part of the model, which would be essential for interpretation. Please clarify.

L295: Please specify what the numbers in brackets refer to.

L318: How can β be equal to one edge of the CI?

L325-328: I wonder if this question could have been answered by including related dyads in the female-male grooming model (L183-198) instead of running an extra test?

L368: Should it not say βdisperser x stable?

As PLOS One is a journal that publishes research of all scientific disciplines, I feel that the focus of this manuscript in its current form might be too narrow to fit. The research questions are indeed very interesting and the results represent an important contribution to our understanding of fission-fusion dynamics in social animals. However, the manuscript currently solely focusses on primates, even though fission-fusion-dynamics and the associated costs and benefits are not unique to primates. Therefore, I would suggest discussing the results and implications in the context of social animals in general, as this would be of great interest for the broad PLOS One readership.

**Do you want your identity to be public for this peer review?** For information about this choice, including consent withdrawal, please see our Privacy Policy

Reviewer #1: No

Reviewer #2: No

Reviewer #3: **Yes: ** Annika Freudiger

---

## [Author Response · Author response to Decision Letter 1]

13 May 2025

Editor Comments

This study analyses affiliation and aggression networks prior, during and after a single group fission event in olive baboons. While previous studies mainly looked at patterns of group fissions, this case study of a male-initiated fission is really interesting, as it provides detailed analyses in the process of group split/s. Hence the study is providing an important foundation for comparative work of group fissions in nonhuman primates. Even though the process seems to be quite unique (male-initiated) from what has been described previously, the results compare well with previous studies, i.e. that females stay with their closest social partners which are close maternal relatives.

The manuscript has been reviewed by three experts and myself. The manuscript is well written and I enjoyed reading it. However, I asked a third reviewer to explicitly evaluate the statistical approach (given the complexity of the analysis). All reviewers raised several important concerns and I agree with them in all points. While the topic is really interesting and the data are sufficient to test the questions raised, the current version of the Ms. has serval serious limitations which I summarize below:

Introduction: would benefit from details on fission types observed in nonhuman primates (apart from male- and female-initiated splits). I think the main pattern are that splits occur across and between matrilines, while mainly females remain with close maternal relatives.

Methods: most reviewer comments are related to the statistical models as they are hard to follow and often less clear. You run a total of eight Bayesian models and five frequentist analyses and it is unclear why you think they are necessary to address your questions. You nicely provide several questions, however, I recommend to link each question with one or more model/s. Furthermore, each analysis should be introduced separated while ideally being numbered (and/or list a short title), so the readers can view the methods and results per model in more detail. You can also think about moving some analytical parts of lower importance into the supplement to increase readability of the main Ms. In the current stage, the analysis seems to have several issues which need to be addressed in order to present it in a way readers can potentially digest it.

Results: the results are difficult to follow. While they are linked to the broader question, they are not link to individual model/s. As pointed out by reviewers, please provide tables with model outcomes for all predictors and random effects including information on sample size and distribution of predictors.

Conclusion: It is recommended to restrict conclusions to the actual results you are presenting.

In the light of these points, a major revision of the Manuscript is needed. If you are willing to provide a complete revision of the Ms, I will be happy to revise my decision and suggest the publication of this Ms. in Plos One. I think the study will be a valuable contribution to the literature.

>> Thank you for the opportunity to revise our manuscript. The manuscript, its results, and its presentation are much improved in light of reviewers’ comments.

Regarding the Introduction, we have revised our description of the two main ‘types’ of fissions. We agree that fissions that preserve vs. break maternal kin relationships are a meaningful distinction, which we now highlight more throughout the Introduction. In general, our intention was to group fissions by their probable proximate triggers (i.e., feeding competition vs. male mating competition) and not by their demographic outcomes. We hope our rewording emphasizes this intended focus.

In the Methods section, we now delineate our models more clearly, using headers that match the ordering of our research questions, as delineated in the Introduction. We have also removed many of the frequentist analyses, provided descriptive data where these would suffice, or moved to Bayesian approaches where we felt these were appropriate (L318-323).

In the Results section, we carry this organizational scheme forward. We also provide complete model output tables in the Supplementary Materials.

Lastly, we have removed the Conclusions section, as it was largely redundant with material provided elsewhere.

Reviewer #1: In this manuscript the authors describe changes in social behavior around a single group-fission event in a population of olive baboons. As a stand-alone case study the analyses will gain value only when others provide similar data fueling future statistical evaluation of the phenomenon. This being said, the manuscript provides a detailed account of the affiliation and aggression network before, during and after the fission event which is rare but desirable. The manuscript would benefit from a thorough revision of the intro where the two types of fission events observed in primates are defined and in the sequence of presentation of material. There is a glitch in the statistical analyses that needs fixing and discussion would benefit from relating the findings to a recent analyses of yellow baboon group fission events relative to female social bond networks. Below I provide comments as they came up during my first reading of the manuscript which mimics the experience of a future reader.

Introduction

Line 42: It may be noteworthy that in marginal habitats chacma baboons have been reported to form one-male groups and that some colleagues believe that these are the conditions during the speciation of chacma baboons that have left a trace in current day social behavior patterns and reproductive strategies including infanticide. As such, chacma baboons would fall within the same group as gorillas and Asian colobines – they have it in them to form one-male groups.

>> We agree that these chacma baboon fission products are quite relevant here. We now elaborate on this point in L46-48.

Line 51: The example does not really illustrate the point that individuals have different goals, because the male priority to recruit a number of females does not have to be at odds with females maintaining their close affiliative relationships if an entire matriline joins the event.

>> We’ve reworded this example to better illustrate this point in L54-56.

Line 56: The reader would benefit from a more detailed description of the published data on group splits in baboons and particularly those that are classified as male initiated. It is not entirely clear which criteria are used to classify events into male vs. female led. Is it the temporal pattern, a male goes first and the females join later? Is it a male actively herding females away from the main group? Is it always only one male or sometimes more than one? Is it not true that any number of females that start a new group will always do so together with one or several males?

>> We’ve revised this passage for clarity. In short, we used this schema to highlight two ‘triggers’ that instigate fission in primates (i.e., female-female feeding competition, and male-male mating competition). while male-driven fissions are those that maximize the instigating male’s reproductive opportunities. However, we note that these categories are not mutually exclusive and inferred by both circumstantial (i.e., the temporal patterning of individual social decisions, the occurrence of male ‘herding’ behaviors) and quantitative lines of evidence (i.e., whether females were able to preserve their kinship ties).

Applied to the current study: What tells us that it was Yohan who led the females and not the females that led Yohan to ENK? Three females and Yohan went to ENK, 1 female and Yohan stayed. The 2 females that went back, retrieved another three to join ENK for a second stint and then decided to do their own thing for which they were joined by Yohan. The story could be told from the perspective of those two females that checked out opportunities, then mustered supporters to join ENK which failed and motivated these females to start a new group.

>> We now provide a more detailed timeline at the beginning of the Results section. Specifically, Yohan first moved into ENK along with three females on January 2015. Thus, at this temporal scale, we cannot say which preceded the other. We now note this in L297-304.

Methods

Kinship: It is not clear whether kinship was modeled as a continuous, dichotomous or a multinomial variable one and how this may have affected the results. On line 291 it seems that relatedness was classified into close kin versus non-kin, i.e. two categories? The data on github suggest a more fine-grained measure.

>> We now clarify this fine-grained measure and provide a more succinct description in L137-140.

It seems that the statistical models do not include random slopes (e.g. the effect of group may vary across subjects or the effect of period may vary across subjects) and their correlations with random intercepts which can drastically increase type 1 error rate. Barr et al 2013 Random e�ects structure for con�rmatory hypothesis testing: Keep it maximal. J Memory and Language, 68:255–278; Schielzeth& Forstmeier 2009 Conclusions beyond support: overcon�dent estimates in mixed models. Behav Ecol 20:416–420.

>> We agree in principle that including random slopes would allow us to determine whether the effects of co-residence and eventual outcome (i.e., our two key predictors) vary across study subjects. However, given the broader complexities of our modeling approach (i.e., multi-membership random intercepts, hurdle gamma models which estimate two parameters for each predictor), random slopes did not converge for our primary models focused on dyadic grooming (i.e., Models 1a-f). We now state this in L195.

Likewise, the key metrics of interest in our other models (i.e., dispersal status, yes/no) did not vary within individuals. Thus, random slopes would not be informative in these cases.

Results

During my first reading I was not entirely sure about how females were assigned to a group. In Fig. 3 the estimated grooming time per hour for individuals that never stayed together clearly differs from zero and is just about ¼ of what you get if individuals lived in one group throughout. Where do you draw the line if there is so much grooming between members of different groups? This is even more extreme for between-sex grooming. Is that typical for the population, the species, baboons in general to see member of different groups grooming regularly? Why are all posterior estimates in the text negative?

>> Group membership was determined during routine observation censuses, and individuals were assigned a monthly group membership based on the majority of these assessments. We state this outright in L95-97.

This is certainly atypical for baboon groups but likely stems from a recent history of fission. We reiterate this point in L68 and L485-489.

As a reader I would prefer to see a table with model outcomes for all predictors and random effects including information on sample size and distribution of predictors (e.g. number of close kin and non-kin dyads).

>> We now provide a thorough set of model output tables in our Supplementary Materials.

Line 295: Something is missing here to indicate which estimate is for which of the groups compared. From the text is looks like dyads that stayed and those that separated were compared with some Wilcoxon Test. Since these dyadic data include repeated measurements the data points are not independent and we would look at a case of pseudo-replication inflating Type I error.

>> This is a fair concern. The initial analysis was meant to serve as a post hoc check on whether Yohan’s relationships predicted these choices in a more targeted fashion. To alleviate these concerns, we’ve removed many of these post hoc analyses, included simple descriptions of these data where they remain useful, or switched to Bayesian approaches that capture data interdependence where we feel this approach would be both productive and applicable.

Line 296: This needs more explanation to make the analysis more accessible in the results section. How can there be as much grooming between females that currently live in the same and in different groups? Why are females grooming outside their group so much? Who is compared to whom here? The issue applies to analyses of female-male grooming and of aggression. Between-group grooming is mentioned in the methods section but appeared to be 10 times less frequent than within (600 vs. 5400 events).

>> Apologies for the confusion, as our original visualization overstated the extent of between-group grooming due to our confounded variables. We also introduce the prevalence of between-group grooming earlier in the Introduction (L68-70) and provide potential explanations in L486-490.

Line 357: there is a word missing after slightly to indicate whether contact aggression was more or less likely.

>> This has been corrected.

Line 365: Perhaps you want to move this part about grooming between groups up to the beginning of the results section, because as you see from my comments, this amount of BG-grooming is unusual and very confusing to read about. The discussion talks about a priori knowledge on BG-grooming since 2011 which is info that could be moved all the way to the introduction.

>> Same as above.

Line 400: It is unclear how this test was run and how data were pseudo-replicated here as indicated by the very large W-value. This would become immediately clear if the sample sizes were reported with the test statistic and p-value.

>> Due to other reviewer concerns, we have removed this analysis and opted to simply state that many of the dispersing females belonged to a single matriline.

Line 409: The conclusion that females were influenced by Yohan and their female kin seems a bit overstated. What is the evidence that rules out that females made their own decisions and Yohan and (and a bunch of subadult males) just joined the move? If as stated here, Yohan was natal to PNG, his natal dispersal had to come at some point anyways and thus may not have been driven by male-male competition but by incest avoidance.

>> We now state outright that inbreeding avoidance is the likely explanation for why Yohan left his natal group, PHG (L459-460).

The conclusion paragraph is redundant with the introduction and can be cut from the manuscript.

>> Given these redundancies, we have removed this paragraph.

It would be great to discuss the results in relation to the Better baboon break-ups analyses by Lerch et al (a study cited here).

>> This is a useful study to reference here. We now elaborate on these data in comparison to ours in the Discussion (L464-466).

Reviewer #2

This was a detailed look at a group fission event, which provided insight into possible factors that influenced decision-making with regards of staying in a group, or leaving it. It is a nice manuscript to read, and I certainly feel I learned a lot about fissions and factors influencing who stays together.

However, I feel the paper would have benefited from more anecdotal information about the event, as this is primarily a case study of a fission. For example, what is missing is also more descriptive information about the initial dispersal of Yuhon. Once he dispersed, did the females immediately follow? Was there a brief period when the females stayed in their group (PHG) before following Yuhon? The female that stayed with Yuhon throughout Dispersal 1 and Dispersal 2, was there something different about her relationship to Yuhon or her female kin in PHG? What was different in the circumstances of the 1 female that did not remain with Yuhon?

>> We now provide a more detailed timeline at the beginning of the Results section (L297-304). Unfortunately, due to the coarse timescale at which group membership is determined in this study population, we cannot say whether there was a ‘lag’ between Yohan’s dispersal and the three females’ dispersal. We also note that the female who remained in ENK throughout the remainder of the study had kin within this group, which may have contributed to her social decisions (L300-302).

The main take-home in the end seems to be that prior mating/sir

---

## [Decision Letter · Decision Letter 1]

4 Jul 2025

Dear Dr. Feder,

The revision was seen by all three previous reviewers. They all agree that the Manuscript improved substantially. Two reviewers still have some smaller clarification issues or ask for visualization of the posterior distributions of model estimates (which I agree would be very helpful to understand your results better) and my comments have also been incorporated well. However, the third reviewer still has an analytical issue. The question is: do you go with those you got along well in the past OR do you prepare for a group fission by deepening certain relationships? These might be two different things assuming that monkeys could anticipate an upcoming fission event. It may also be difficult to analyze all these phases of this complex process in one model, because these females may have different problems to solve in each phase, e.g. who to join when a fission occurs or who to groom in a new group for successful integration. I think the reviewer may have a point here, which was not really clear in the original version. Therefore, I need to ask you to address these concerns before I can suggest this Ms. for publication. I decided to give you another major revision as this will provide you with more time, however, you can of course submit earlier.

Two smaller comments:

L 195: you used no random slopes at all? Usually you still include all identifiable random slopes if your random correlations cause convergence issues. Pls specify.

L 219: related m-f dyads, you mean mother-son or father-daughter dyads and maternal siblings? Pls specify.

We look forward to receiving your revised manuscript.

Kind regards,

Anja Widdig

Academic Editor

PLOS ONE

Reviewers' comments:

Reviewer's Responses to Questions

**Comments to the Author**

Reviewer #1: (No Response)

Reviewer #2: All comments have been addressed

Reviewer #3: (No Response)

2. Is the manuscript technically sound, and do the data support the conclusions?

Reviewer #1: No

Reviewer #2: Yes

Reviewer #3: Yes

3. Has the statistical analysis been performed appropriately and rigorously?

Reviewer #1: No

Reviewer #2: Yes

Reviewer #3: I Don't Know

4. Have the authors made all data underlying the findings in their manuscript fully available?

Reviewer #1: Yes

Reviewer #2: Yes

Reviewer #3: Yes

5. Is the manuscript presented in an intelligible fashion and written in standard English?

Reviewer #1: Yes

Reviewer #2: Yes

Reviewer #3: Yes

Reviewer #1: I came away from reading the revised version with a better understanding of the analytical strategy and new concerns about its appropriateness.

Firstly, the DAGs provided in the supplement scrutinize the causal hypotheses underlying the statistical models. I do not share the view that grooming relationships are a causal consequence of fission outcome. There is no theoretical reason or empirical work suggesting that females form closer relationships with particular females in anticipation of an upcoming fission event. These grooming relationships form whether the group fissions or not. And when it is time to decide which daughter group to move to, females go with their close partners or the majority of their close partners, i.e. the causality is reversed and grooming drives fission outcome. The causal relationship between co-residence and grooming relationship is straightforward: if we do not live together, we do not see each other and cannot groom. In the study population groups seem to aggregate enough to provide some, but surely not equal grooming opportunity for those living in the same and different groups. The causality between co-residence and fission outcome is less clear.

Secondly, the way data are analyzed across groups may be confounded by very strong co-residence effects that blur the interesting facts. If the 2-3 groups are treated as one large network as suggested by the network graphs, all analyses will reproduce who is living in which group in each study period. Every dyad was labeled as one of the four fission outcome classes and their grooming relationship measured four times, once per study period. Dyads that always stay together necessarily have a stronger grooming relationship across the four periods than other dyads, because members of the other dyads did not live in the same group in at least some of the periods (666 out of 806 values for dyads from different groups are zero compared to 317 out of 715 from the same group). If a dyad lived together only briefly, then this brief period is when their grooming should have been measured. The way the data are analyzed does not tell us how the dispersing females did or did not integrate into the grooming network of ENK during periods 2 and 3, or why one female stayed in ENK while other dispersers did not. In the current analyses, this information is now hidden behind very strong patterns driven by co-residence.

I list my other comments as they came up during my first reading of the revised ms.

Please note that the line numbers I use here refer to the manuscript version with tracked changes.

Line 12: Please consider rephrasing. It does not seem to be adequate to say here that females followed subsequently if the male went at the same time as the females. On line 123 and in the response letter it sounds like the male went TOGETHER with three females and on line 381 of the results section it is clearly stated that it is unknown whether the male went first.

Line 15: Together with the next sentence this is difficult to parse for a first time reader. Is the sentence only about females that remain, i.e. do not disperse, or also about the females that did disperse and did so with their main female grooming partners. The same ambiguity applies to the co-residents in the next sentence. Are co-residents the individuals that did not disperse or those that end up in the same group at any step throughout the dispersal event? In the abstract it may be useful to use one word for one thing to better control what the reader takes home.

Line 45: There is no first to this “Second, …”

Line 50: The references do not seem to support the statement here. Langurs in Jodhpur (Winkler et al.) live in one-male groups and are subject to take-over from bachelor groups and the Thomas langurs Sterck studied disintegrate when the single resident male cannot monopolize it any more and females move to other groups or together form a new group with a former bachelor male. Both situations seem to be fundamentally different from a multi-male group fission as described for baboons where individual in the daughter groups had been living together for a while before the split.

Line 53: There is no reference provided for cases of aggressive herding. Does this occur in gorillas/langurs, in baboons or in both groups with male-initiated fissions?

Line 56: I the connecting phrase likewise helpful here? You are sending the reader on a quest for what is similar between the two sentences without making it explicit.

Line 63: What are THESE two fission patterns? The breaking kin bonds vs. not OR female vs. male initiated? Be explicit.

Line 86: What is the difference between dispersal patterns and fission outcomes? These terms have not been defined and are therefore ambiguous.

Line 88: It is not clear why and how grooming network structure should shift prior to group fission, because the intro does not prime the reader.

Line 93: The topic of eviction and targeted aggression is not properly introduced. A reader may also expect to see targeted aggression to be contrasted with overall elevated aggression among females possibly resulting from increased feeding competition triggering female led fission.

Line 97: No predictions or background are provided about inter-group grooming except for mentioning that there is a lot of it.

Line 186: The sequence of sections is a bit odd, because we read about how the data were analyzed for dominance relationships before we learn how data were acquired.

Line 390: From the literature reviewed in the intro, it makes a lot of sense to ask whether parallel dispersal involved females with strong grooming relationships. So I understand why one would model grooming in period 1 as a function of final fission outcome, even if the causality is reversed here. What I do not understand is why this analysis is conflated with between group grooming which ought to be less frequent than within group interactions or the definition of group membership is off. Is it not self-evident that females that always stayed together had the strongest relationships in these models? I am looking across four periods that would include for everybody else 1-3 periods where partners did not co-reside which must reduce the opportunity to groom. It seems that the models of eventual outcome are just a different way of modelling co-residence in four steps instead of binary (the same goes for male-female relationships).

Line 242: Apart from the conceptual issue raised about Line 390ff, the paper would be much lighter if only the models with both predictors would be maintained. It is the best model for males and not worse than individual models for females. One way to assess how severely the covariation among predictors may have affected model outcomes is to use a frequentist glmm with the same model structure and investigate variation inflation factors. But see my comments re Line 390ff.

Figure 2: The data in figure 2 (formerly Fig. 3) look very different between the two versions of the manuscript. Can the authors explain how all differences became so much more pronounced in the revised version when both the legends say the same which suggest the same data have been plotted?

Line 401: Like the dispersal outcome analyses in Model 1, the analysis of maternal relatedness and grooming across all periods does not seem very informative, because it just replicates co-residence effects. The interesting part is, whether maternal relatedness in period 1 predicted who ended up in different groups.

Line 423: An analysis of grooming vis-à-vis siring across all periods only makes sense, if males sire a lot of offspring outside their social group – otherwise the analysis will just reproduce co-residence patterns. In contrast, information on Yohan’s siring success with the dispersing versus non-dispersing PHG females makes a lot of sense.

Line 443: Is it not the grooming networks that became denser instead of the groups themselves.

Did the grooming network analyses include subadult males? All other analyses seem to concern only fully adult individuals, but the laipikia_dispersal data frame has subadult males with network metrics.

Line 521: Why is groups plural here? All 6 females that dispersed came from one group, right? Does it follow that only data from period 1 were analyzed to compare whether dispersing females were targeted for aggression which then led to dispersal in period 2 and 3? Or is it a question whether the females that went back to PHG had been targeted by ENK residents which triggered their (temporary) return home where they may have been targeted again leading to their permanent emigration? In other words: do these six females play the same social role across periods 1-3 and of not, does it make sense to model aggression received across all periods?

Line 559: It seems that the out-group grooming comparison is a bit unfair, because opportunity to groom outside the group increased in period 4 (fission) when the six dispersers gained many more out-group partners than they had in any other constellation.

Line 563: It seems odd to say that fm out-group grooming was disproportionately directed from females to males when in fact all the many dyads involving a disperser male must have involved Yohan whereas a similar number of dyads in the fm category came from six dispersed females.

Line 586: Is the plural correct here, are there several males that had sired the dispersing females’ last offspring or is it all only about Yohan? If it is the former, because the events unfolded over such a long time, we need more details.

Reviewer #2: I have read through the manuscript and the response to reviewers. In my opinion the authors have addressed the reviewer comments adequately.

A few minor comments:

Line 348: Seems like Yuhon did just find avoiding inbreeding before leaving PHG. In macaques, natal males from high-ranking matrilines can do very well without leaving. It could also be that Yuhon followed the first three dispersing females in order not to lose his access to preferred mates (ones he had already successfully bred with). Yes, Yohan reduces his inbreeding risk by dispersing, but the female-first dispersal scenario cannot be ruled out. This section is written as support for siring histories shaping female social decisions, but the same argument can be made for siring histories shaping male dispersal decisions in this case.

Lines 355-356: “the dispersal of PHG females into ENK may have triggered social instability”. Maybe I don’t understand modularity well enough, but isn’t this just the result of the new arrivals not integrating well into the ENK. Does this really count as “social instability”? I would argue not, if the original ENK group members didn’t themselves become more modular in their associations. Maybe in connection with the increase in aggression does the “social instability” comment make more sense, but here it seems a bit out of place.

Line 358-360: Is this supposed to say either “female-male grooming” or "male-female grooming", instead of “83.8% of between-group grooming”?

Line 443: Was the female that stayed in ENK not a member of the matriline? If so, it would be good information to state. An earlier sentence (line 301) mentions that she had kin ties with ENK.

Figure S1.

- Describe what the colors/shading represent.

- “Co-residence” in a) is the same as “current membership” in b? If so, why different labels? It only introduces confusion.

- “co-residence likely facilitated past siring opportunities, while a prior siring history may compel males to have remained in that social group.” I am confused about this statement. If you change past siring history, then you can change an individual's decision about group membership (you removed a reason to stay). But, if you change an individual's current group membership, you don’t change the prior siring history. Please clarify.

The authors don’t mention whether they checked trace plots to see if each chain was ‘behaving well’.

Authors don’t mention if they ran prior predictive checks (typically in brms: pp_check()).

Reviewer #3: The editor asked me only to review the statistical methods. This section of the manuscript has considerably improved, and the applied statistical tests are now much clearer explained. I still have a few comments:

L181: Please specify: which contain many zeroes in the response variable

L186-190: I would suggest moving this explanation to the results section.

L194: Maybe “dyadic” instead of “interaction”?

L200: Mention the response variable you used (grooming duration measured in minutes?)

L219-220: Please add how many dyads were omitted.

L231-243: This is an interesting approach to overcome the issue with differences in sampling effort. However, I am unsure if the posterior predictions of the two models (female-female, female-male) can be pooled. The models were run with a different set of predictors (mat relatedness and rank difference for same sex, male rank and recent common offspring for different sex). They might capture different relationships between the predictors and the response. Pooling these predictions essentially assumes that the effects of the main predictor (co-residency, eventual outcome) are directly comparable across these two models, despite the different context set by the control predictors. Please elaborate on this.

L247-248 and L276-280: Please report the percentage of zeroes in the response variable here as well.

L290-294: The authors report that they ran the models in brms with 4 chains with 2,000 iterations each, which is the default setting for brms. Of these 2,000 iterations per chain, by default brms runs 1,000 iterations each to warm-up and 1,000 iterations to sample. As the authors did not specify the number of warm-up and sampling iterations used, I assume they applied default settings. However, in the supplementary, they report bulk and tail ESS of well over 4,000 for most parameters. To my understanding, such high ESS values are only possible when >> 2,000 iterations were used. Please clarify which settings you used for which model.

L292: Why was 500 used as a threshold for bulk and tail ESS? The default of brms is ~ 1,000 to consider an estimate stable. Also, your reported results exceed 500 by far. Please clarify.

L293-294, L186-190: Even though the authors apply Bayesian statistics, they rely on a certain threshold to differentiate between biologically meaningful and non-meaningful results. However, it is advised not to stick to arbitrarily chosen thresholds to assess biological relevance in a Bayesian approach (even though 89% is widely used, its initial choice was arbitrary; McElreath 2020, Statistical Rethinking). E.g. the authors report that rank differences did not predict grooming relationships (L319-320), even though the zero is very close to the upper boundary of the CI. Based on these numbers, I would argue that there might be a pattern, just not a strong one (i.e., a “trend” in frequentist terms), although one would need to see the whole posterior distribution for proper interpretation. Hence, I think the manuscript would benefit a lot from adding visualisations of the posterior distributions of model estimates. This would make it much easier to compare the position and breadth of distributions among each other and get a feeling for the certainty and biological meaning of the estimates.

**Do you want your identity to be public for this peer review?** For information about this choice, including consent withdrawal, please see our Privacy Policy

Reviewer #1: No

Reviewer #2: No

Reviewer #3: No

---

## [Author Response · Author response to Decision Letter 2]

16 Aug 2025

Editor Comments

The revision was seen by all three previous reviewers. They all agree that the Manuscript improved substantially. Two reviewers still have some smaller clarification issues or ask for visualization of the posterior distributions of model estimates (which I agree would be very helpful to understand your results better) and my comments have also been incorporated well. However, the third reviewer still has an analytical issue. The question is: do you go with those you got along well in the past OR do you prepare for a group fission by deepening certain relationships? These might be two different things assuming that monkeys could anticipate an upcoming fission event. It may also be difficult to analyze all these phases of this complex process in one model, because these females may have different problems to solve in each phase, e.g. who to join when a fission occurs or who to groom in a new group for successful integration. I think the reviewer may have a point here, which was not really clear in the original version. Therefore, I need to ask you to address these concerns before I can suggest this Ms. for publication. I decided to give you another major revision as this will provide you with more time, however, you can of course submit earlier.

>> Thank you for these comments, which have pointed us towards a more thorough explanation of our objectives in the introduction. Our aim in modeling social relationships across these four time periods was not to see whether individuals that stayed together deepened their relationships per se but rather to see (i) whether individuals that split apart weakened their relationships in the period prior to their departure and (ii) whether the dispersing individuals from PHG strengthened their intergroup grooming relationships with ENK residents in anticipation of their dispersal (i.e., whether they engaged in ‘shopping’ before their dispersal). Although we had considered modeling this sequence via two separate models (i.e., first the females’ initial dispersal using dyads from PHG, second the dispersing females’ failure to integrate using dyads from ENK), we ultimately opted for this complex model in order to detect these subtle patterns. We’ve revised the text in the Introduction (L70-73) and Methods (L200-206) to make this reasoning and framework clearer, adding references to previous research suggesting that primates prune their social ties in the years leading up to fission.

That being said, the lack of support for interaction effects makes it clear that this was not the case. In other words, individuals did not shift their social preferences as they faced different social problems across each sampling period, and individuals did not subtly begin to groom beyond their social group in anticipation of their dispersal. Instead, individuals simply stayed with those they got along with in the past. We now comment on this result and its implications more explicitly in the Discussion in L465-468.

Two smaller comments:

L 195: you used no random slopes at all? Usually you still include all identifiable random slopes if your random correlations cause convergence issues. Pls specify.

>> We now clarify that we did not use random slopes for our primary factors of interest (e.g., sampling period, eventual outcome) because there were not multiple observations for each category for each individual (L191-193). For instance, some females were not present during specific periods (because they disappeared or matured midway through the study) and not all eventual outcome types were available for each female (by definition, females that never dispersed did not belong to any briefly together dyads), and thus we were unable to reliably estimate random slopes for these categorical variables.

L 219: related m-f dyads, you mean mother-son or father-daughter dyads and maternal siblings? Pls specify.

>> This is a fair point for clarification, as we only know maternal kin from demographic records, and genetic sampling is relatively shallow (i.e., we may have inadvertently included unconfirmed father-daughter dyads). We’ve modified the text to clarify that we only removed maternal kin and now provide sample sizes for the omitted maternal kin dyads in L235-236.

Reviewer #1: I came away from reading the revised version with a better understanding of the analytical strategy and new concerns about its appropriateness. Firstly, the DAGs provided in the supplement scrutinize the causal hypotheses underlying the statistical models. I do not share the view that grooming relationships are a causal consequence of fission outcome. There is no theoretical reason or empirical work suggesting that females form closer relationships with particular females in anticipation of an upcoming fission event. These grooming relationships form whether the group fissions or not. And when it is time to decide which daughter group to move to, females go with their close partners or the majority of their close partners, i.e. the causality is reversed and grooming drives fission outcome. The causal relationship between co-residence and grooming relationship is straightforward: if we do not live together, we do not see each other and cannot groom. In the study population groups seem to aggregate enough to provide some, but surely not equal grooming opportunity for those living in the same and different groups. The causality between co-residence and fission outcome is less clear.

>> Thank you for this comment. We agree that grooming relationships are not a causal outcome of fission and the DAG presented in the supplementary materials obfuscated this presumed relationship (i.e., that demographic outcomes following fission are the consequence of grooming relationships). Our intention in using fission outcome as a predictor was to capture the individuals’ latent social preferences (which are unmeasurable). In other words, current group membership is a reflection of the females’ past social preferences, while eventual outcome is a reflection of the females’ ongoing social preferences. This eventual outcome categorization may account for variation above and beyond that associated with current group membership. As far as how co-residence is linked to fission outcome, we merely meant to indicate that individuals who are co-resident are more likely to end up in the same groups than individuals who are not co-resident.

We now comment on these causal relationships more explicitly in Figure S1 and its caption and provide more reasoning for this approach throughout the Main text.

Secondly, the way data are analyzed across groups may be confounded by very strong co-residence effects that blur the interesting facts. If the 2-3 groups are treated as one large network as suggested by the network graphs, all analyses will reproduce who is living in which group in each study period. Every dyad was labeled as one of the four fission outcome classes and their grooming relationship measured four times, once per study period. Dyads that always stay together necessarily have a stronger grooming relationship across the four periods than other dyads, because members of the other dyads did not live in the same group in at least some of the periods (666 out of 806 values for dyads from different groups are zero compared to 317 out of 715 from the same group). If a dyad lived together only briefly, then this brief period is when their grooming should have been measured. The way the data are analyzed does not tell us how the dispersing females did or did not integrate into the grooming network of ENK during periods 2 and 3, or why one female stayed in ENK while other dispersers did not. In the current analyses, this information is now hidden behind very strong patterns driven by co-residence.

>> We agree that modeling these two correlated variables creates problematic complexities due to confounding. These concerns motivated our choice during the previous revision to construct and compare models including (i) co-residence alone; (ii) eventual outcome alone; and (iii) co-residence and eventual outcome combined to determine whether these two variables capture non-redundant information within the dataset. While modeling co-resident dyads alone would be a more focused representation of the fission process, we chose the current approach because we wanted to know whether females engaged in ‘shopping’ prior to their dispersals (i.e., whether they increased the amount of grooming of outgroup individuals before making the decision to leave). To capture these potential temporal patterns, we included interaction terms between study period and the eventual outcome categories. However, the interaction term did not improve model fit (in models that both did and did not include co-residence as an additional term), which suggests that these temporal changes did not occur. In other words, dispersing females did not integrate into the broader group, as the “briefly together” dyads formed weak relationships across all of the sampling periods (even during their periods of overlap). Likewise, dyads that split apart did not weaken their relationships in anticipation of their split.

I list my other comments as they came up during my first reading of the revised ms.

Please note that the line numbers I use here refer to the manuscript version with tracked changes.

Line 12: Please consider rephrasing. It does not seem to be adequate to say here that females followed subsequently if the male went at the same time as the females. On line 123 and in the response letter it sounds like the male went TOGETHER with three females and on line 381 of the results section it is clearly stated that it is unknown whether the male went first.

>> Thank you for this suggestion. We’ve modified the text in L11-12 to avoid this false impression.

Line 15: Together with the next sentence this is difficult to parse for a first time reader. Is the sentence only about females that remain, i.e. do not disperse, or also about the females that did disperse and did so with their main female grooming partners. The same ambiguity applies to the co-residents in the next sentence. Are co-residents the individuals that did not disperse or those that end up in the same group at any step throughout the dispersal event? In the abstract it may be useful to use one word for one thing to better control what the reader takes home.

>> We’ve modified the text to be more consistent and more closely match our modeling approach.

Line 45: There is no first to this “Second, …”

>> Thanks for pointing this out. We’ve removed this.

Line 50: The references do not seem to support the statement here. Langurs in Jodhpur (Winkler et al.) live in one-male groups and are subject to take-over from bachelor groups and the Thomas langurs Sterck studied disintegrate when the single resident male cannot monopolize it any more and females move to other groups or together form a new group with a former bachelor male. Both situations seem to be fundamentally different from a multi-male group fission as described for baboons where individual in the daughter groups had been living together for a while before the split.

>> This is a fair concern regarding the comparability across species and sites. We’ve added text to clarify that many of these fissions occur when male takeovers divide a social group into multiple one-male groups. Our intention in referring to these taxa was to fold these examples into our broader definition of fission, in which we consider any demographic scenario in which one social group divides into multiple daughter groups. Also, the Sterck and Korstjens citation refers to a book chapter that discusses multiple cases where groups divided into new daughter groups and females are presented with the opportunity to join a new male or remain with a familiar male (including examples from not only colobines and gorillas but also chacma baboons, etc.). This broadly mirrors the scenario described for chacma baboons in the following sentence (L47-49).

Line 53: There is no reference provided for cases of aggressive herding. Does this occur in gorillas/langurs, in baboons or in both groups with male-initiated fissions?

>> We now clarify that this is pertaining to baboons in L49-50.

Line 56: I the connecting phrase likewise helpful here? You are sending the reader on a quest for what is similar between the two sentences without making it explicit.

>> We’ve removed this connecting phrase for clarity.

Line 63: What are THESE two fission patterns? The breaking kin bonds vs. not OR female vs. male initiated? Be explicit.

>> We now clarify our meaning in L56.

Line 86: What is the difference between dispersal patterns and fission outcomes? These terms have not been defined and are therefore ambiguous.

>> We agree that this was ambiguous, as they largely pertain to the same process, and have simplified the wording in L75-76.

Line 88: It is not clear why and how grooming network structure should shift prior to group fission, because the intro does not prime the reader.

>> We’ve added some text and provided citations demonstrating that social networks often change prior to fission in L38-39. We’ve also amended our research questions to make it clearer why we modeled grooming relationships across all of these periods.

Line 93: The topic of eviction and targeted aggression is not properly introduced. A reader may also expect to see targeted aggression to be contrasted with overall elevated aggression among females possibly resulting from increased feeding competition triggering female led fission.

>> We’ve simplified the text here to clarify the distinct possibilities and prime the readers for our analytical approach in L79-80.

Line 97: No predictions or background are provided about inter-group grooming except for mentioning that there is a lot of it.

>> We’ve added a sentence to make it clear that intergroup grooming is uncommon for baboons and clarify why we included these data in L70-73.

Line 186: The sequence of sections is a bit odd, because we read about how the data were analyzed for dominance relationships before we learn how data were acquired.

>> We’ve re-ordered these paragraphs for clarity.

Line 390: From the literature reviewed in the intro, it makes a lot of sense to ask whether parallel dispersal involved females with strong grooming relationships. So I understand why one would model grooming in period 1 as a function of final fission outcome, even if the causality is reversed here. What I do not understand is why this analysis is conflated with between group grooming which ought to be less frequent than within group interactions or the definition of group membership is off. Is it not self-evident that females that always stayed together had the strongest relationships in these models? I am looking across four periods that would include for everybody else 1-3 periods where partners did not co-reside which must reduce the opportunity to groom. It seems that the models of eventual outcome are just a different way of modelling co-residence in four steps instead of binary (the same goes for male-female relationships).

>> We now clarify in the Introduction that our aim in including these between-group ties was to examine whether the dispersing females engaged in “shopping” prior to their dispersal. In other words, did they start grooming with ENK females at higher rates before they decided to leave their natal group.

Line 242: Apart from the conceptual issue raised about Line 390ff, the paper would be much lighter if only the models with both predictors would be maintained. It is the best model for males and not worse than individual models for females. One way to assess how severely the covariation among predictors may have affected model outcomes is to use a frequentist glmm with the same model structure and investigate variation inflation factors. But see my comments re Line 390ff.

>> We agree in principle. However, due to the strong correlation between current group membership and eventual outcomes, solely presenting the models using both terms could inadvertently give the false impression that females do not groom most with the other females from their so

---

## [Decision Letter · Decision Letter 2]

15 Sep 2025

Social patterns underlying a new group formation in olive baboons

PONE-D-25-06688R2

Dear Dr. Feder,

We’re pleased to inform you that your manuscript has been judged scientifically suitable for publication and will be formally accepted for publication once it meets all outstanding technical requirements.

Kind regards,

Brittany N. Florkiewicz, Ph.D.

Academic Editor

PLOS ONE

Additional Editor Comments (optional):

Reviewers' comments:

Reviewer's Responses to Questions

**Comments to the Author**

Reviewer #1: All comments have been addressed

Reviewer #2: All comments have been addressed

2. Is the manuscript technically sound, and do the data support the conclusions?

Reviewer #1: Yes

Reviewer #2: Yes

3. Has the statistical analysis been performed appropriately and rigorously?

Reviewer #1: Yes

Reviewer #2: Yes

4. Have the authors made all data underlying the findings in their manuscript fully available?

Reviewer #1: Yes

Reviewer #2: Yes

5. Is the manuscript presented in an intelligible fashion and written in standard English?

Reviewer #1: Yes

Reviewer #2: Yes

Reviewer #1: I am still not fully convinced of the statistical approach that so heavily rests on a complex interaction term to capture fundamentally different social situations. I leave it to the reader to make up their own mind about that. The manuscript provides all infomration neccessary to do so.

I have no further comments.

Reviewer #2: The manuscript is much improved, as the authors have done a thorough job in their revision. There is improved clarity with added detail about the progression of the fission, and the individuals involved. The parameters of the models are also better detailed.

**Do you want your identity to be public for this peer review?** For information about this choice, including consent withdrawal, please see our Privacy Policy

Reviewer #1: No

Reviewer #2: No

---

## [Editor Report · Acceptance letter]

PONE-D-25-06688R2

PLOS ONE

Dear Dr. Feder,

I'm pleased to inform you that your manuscript has been deemed suitable for publication in PLOS ONE. Congratulations! Your manuscript is now being handed over to our production team.

Kind regards,

on behalf of

Dr. Brittany N. Florkiewicz

Academic Editor

PLOS ONE